# A Small Target Forest Fire Detection Model Based on YOLOv5 Improvement

Zhenyang Xue [1], Haifeng Lin [1,*] and Fang Wang [2,*]

1    College of Information Science and Technology, Nanjing Forestry University, Nanjing 210037, China
2    College of Electronic Engineering, Nanjing XiaoZhuang University, Nanjing 210038, China
*    Correspondence: haifeng.lin@njfu.edu.cn (H.L.); wangfang0182217@njxzc.edu.cn (F.W.);
     Tel.: +86-25-8542-7827 (H.L.); +86-25-86175539 (F.W.)

**Abstract:** Forest fires are highly unpredictable and extremely destructive. Traditional methods of manual inspection, sensor-based detection, satellite remote sensing and computer vision detection all have their obvious limitations. Deep learning techniques can learn and adaptively extract features of forest fires. However, the small size of the forest fire target in the long-range-captured forest fire images causes the model to fail to learn effective information. To solve this problem, we propose an improved forest fire small-target detection model based on YOLOv5. This model requires cameras as sensors for detecting forest fires in practical applications. First, we improved the Backbone layer of YOLOv5 and adjust the original Spatial Pyramid Pooling-Fast (SPPF) module of YOLOv5 to the Spatial Pyramid Pooling-Fast-Plus (SPPFP) module for a better focus on the global information of small forest fire targets. Then, we added the Convolutional Block Attention Module (CBAM) attention module to improve the identifiability of small forest fire targets. Second, the Neck layer of YOLOv5 was improved by adding a very-small-target detection layer and adjusting the Path Aggregation Network (PANet) to the Bi-directional Feature Pyramid Network (BiFPN). Finally, since the initial small-target forest fire dataset is a small sample dataset, a migration learning strategy was used for training. Experimental results on an initial small-target forest fire dataset produced by us show that the improved structure in this paper improves mAP@0.5 by 10.1%. This demonstrates that the performance of our proposed model has been effectively improved and has some application prospects.

**Keywords:** forest fire detection; YOLOv5; BiFPN; CBAM; transfer learning

## 1. Introduction

As an important part of terrestrial ecosystems, forests not only provide a variety of forestry products, but also have an irreplaceable role in regulating ecological balance and provide powerful economic and ecological benefits. However, forest fires are major forest disasters that cause global forest resource loss and human injury and impact forest ecosystem safety. Therefore, it is necessary to research initial forest fire identification.

Forest fires spread rapidly due to rapid convection of air and abundant oxygen in forests. Therefore, it is necessary to detect forest fires at the early stage of their formation. Forest fire detection was first done by manual inspection, but the cost of human and material resources is high, and the efficiency is low. due to this, manual inspections were soon replaced by sensor-based detection. The sensor-based detection system [1–3] works well in small indoor spaces. Among the sensors that are used for forest fire detection are smoke sensors, gas sensors, temperature sensors, humidity sensors, integrated sensors, etc. However, the detection distance is limited, its installation cost is very high, and it also has to face complex communication and power supply networking problems. In addition, the sensors cannot provide important visual information which can help firefighters promptly grasp the situation at the fire scene [4]. Therefore, this method may not be suitable for

large spaces or large areas like forests which are very different from indoor environments. Satellite remote sensing is not only unable to detect small-area fires, but the detection is affected by weather as well as cloud cover. In addition, because satellites only provide a complete image of the Earth every 1–2 days [5], real-time rapid detection is not possible. The high spatial resolution multispectral satellites can not only take clearer satellite images, but some of them can detect early forest fires through short-wave infrared and visible near-infrared bands. However, satellites are still unable to combine temporal and spatial resolution and are currently unable to achieve real-time detection of forest fires. The camera as a sensor has better real-time performance than satellite images. When mounted on a drone, it can also detect forest fires in more remote forests. Using cameras as sensors can help firefighters get a more complete picture of the forest fire scene compared to sensors which do not provide images, such as smoke sensors.

With the rise of computer vision technology, researchers have started to use digital image processing technology for forest fire detection. C. Emmy Prema et al. [6] used YUV color space analysis of smoke for multi-features to detect early forest fires. Pathare, Suyog J. et al. [7] determined the filtering effect of the flame pixels from the intensity of the R component by means of RGB color space and confusion estimation. The experiments show that this method possesses a relatively good application prospect. Hossain et al. [8] used a UAV to capture images of fires, using a special fire color and a multicolor local binary spatial pattern to analyze and detect forest fire flames and smoke from the captured images. Ding et al. [9] proposed an improved flame-recognition color space (IFCS) based on chaos theory and a k-medoids particle swarm optimization algorithm. Khondaker et al. [10] proposed a multi-level fire detection framework which analyzes the color information, shape change and optical flow estimation patterns of fires. In addition, both static and dynamic characteristics are considered to reduce the false alarm rate and computational complexity. In summary, most studies in the field of fire detection methods based on image processing techniques rely on manually extracted features such as colors, shapes, and textures to detect fires.

With the development of hardware and software technology and the enhancement of computer arithmetic, more and more scholars are beginning to study the use of deep learning to detect forest fires. Muhammad et al. [11] propose a cost-effective fire detection CNN architecture for surveillance videos. The early fire detection framework proposed by them uses fine-tuned convolutional neural networks for CCTV surveillance cameras to detect fires in different indoor and outdoor environments. Kinaneva et al. [12] propose a platform that uses unmanned aerial vehicles (UAVs), which constantly patrol over areas potentially threatened by fire. Li et al. [13] proposed novel image fire detection algorithms based on the advanced object detection CNN models of Faster-RCNN, R–FCN, SSD, and YOLO v3. Guan et al. [14] propose a color-attention neural network that consists of repeated blocks of color-attention modules (MCM). Each MCM module is able to extract color feature information from the region. Seyd Teymoor Seydi et al. [15] present Fire-Net, a deep learning framework trained on Landsat-8 imagery for identifying active fires and burning biomass. Sudhakar et al. [16] introduced the multi-UAV system utilized in this examination for agreeable FFD. In conclusion, the researchers proposed multiple network structures to extract image features step-by-step with the help of a multi-layer structure which has higher precision, better robustness and better real-time performance than traditional methods.

In this paper, an improved forest fire detection system based on YOLOv5 [17] is proposed. First, the feature extraction module in the original YOLOv5 cannot extract the effective information of small forest fire targets well because of the small forest fire targets at the early stage of long-distance shooting. Therefore, the original SPPF module of YOLOv5 is modified in this paper. Second, in order to resolve the problem of missing information in YOLOv5 caused by the low number of pixels in small targets of forest fires, the CBAM [18] attention module is added. By using the CBAM attention module, small forest fire targets can be paid attention to, and image features can be recognized more effectively. Third,

by adjusting the Neck layer species PANet [19] structure to the BiFPN [20] structure, the original YOLOv5 model can better balance information at different scales. Last but not least, a transfer learning strategy is used for training, since the initial small-target forest fire dataset is a small sample dataset.

The rest of this paper is organized as follows. In Section 2, the dataset and model evaluation metrics used in the experiments are given, and the structure of the small-target forest fire detection model in this paper is described in detail. Section 3 shows the configuration of the experiments and some of the training parameters settings. In addition, the effects of the CBAM attention module, SPPFP module, and BiFPN on forest fire recognition are experimentally verified. The experimental results are discussed and analyzed in Section 4. Section 5 concludes the entire work.

## 2. Materials and Methods

### 2.1. Datasets

Forest fire detection based on Yolov5 is heavily dependent on the quality of the dataset. Due to this, using high quality datasets in the training process allows deep learners to extract more effective features. First, we obtained images of forest fires not only from the web by writing crawler scripts, but also from some public forest fire datasets. After that, we selected the images that were suitable for training by hand. In order to ensure that our deep learner is able to recognize different kinds of forest fires, we have selected images of tree trunk fires, ground fires, long-distance photography of forest fires, etc. Finally, these images were divided into 3170 forest fire datasets and 150 small-target forest fire datasets. We used these two datasets for transfer learning training. In this article, targets smaller than 32 * 32 pixels are referred to as "small targets" according to the Microsoft coco standard.

The images in the forest fire dataset were taken after a number of large fires that had large flame areas and dark flames. Representative samples of the forest fire dataset are shown in Figure 1.

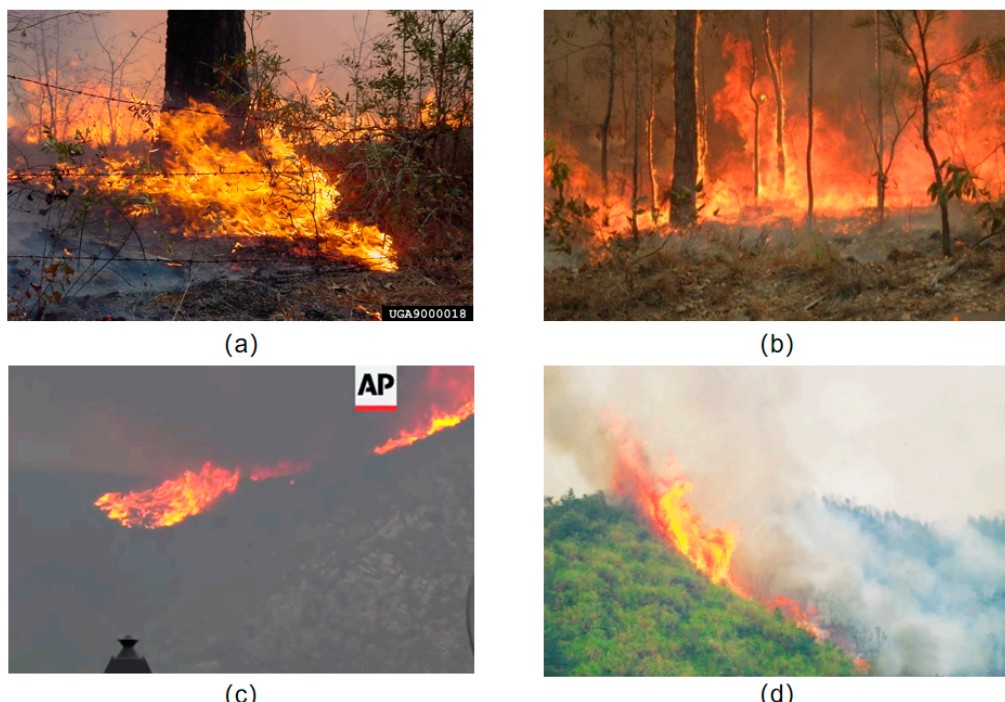

**Figure 1.** Representative forest fire images from the forest fire dataset: (**a**) ground fire, (**b**) canopy fire, (**c**) remote shot of a forest fire, and (**d**) remote shot of a forest fire.

The small-target forest fire datasets are initial forest fire photos, long-range photography of forest fires, and UAV overhead photography of forest fires with small flame areas. Representative samples of the small-target forest fire dataset are shown in Figure 2.

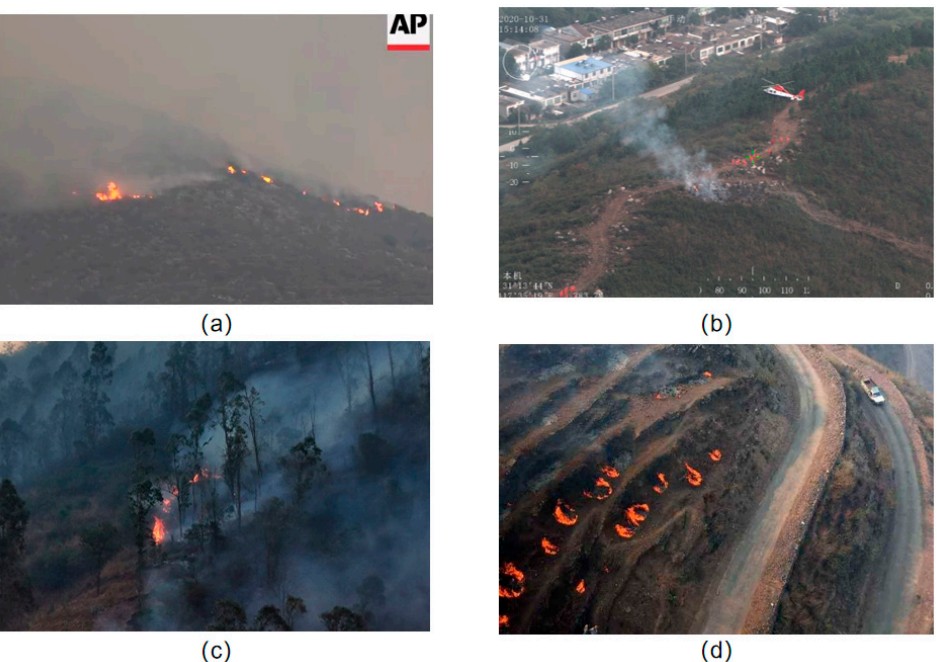

**Figure 2.** Representative forest fire images from the small-target forest fire dataset: (**a**) long-range photography of forest fires, (**b**) UAV overhead photography of forest fires, (**c**) initial forest fire photos, and (**d**) UAV overhead photography of forest fires.

*2.2. YOLOv5*

YOLO is a fast and compact object detection model. In comparison to other networks of the same size, YOLO offers superior performance. YOLOv5 is an advanced target detection network model introduced by the Ultralytics LLC team. YOLOv5 has a faster inference time and a higher detection accuracy than the YOLOv4. In addition, the YOLOv5s module has a small memory footprint, making it easy to deploy in subsequent practical use.

YOLOv5 is divided into four parts: Input, Backbone, Neck, and Output. The forest fire detection model proposed in this paper is based on YOLOv5s in version 6.1 of YOLOv5. The structure of the YOLOv5 model in version 6.1 is shown in Figure 3. First, the input contains Mosaic data augmentation, adaptive anchor frame calculation, and adaptive image scaling. By randomly scaling, randomly cropping, and randomly arranging four images, mosaic data augmentation dramatically improves the dataset sample and strengthens the network. Second, CSP1 and CSP2 are designed with reference to CSPNet [21] and have two different Bottleneck CSP structures. The purpose is to reduce redundant information. As a result, the parameters and FLOPS of the model are reduced, not only to increase the speed and accuracy of inference, but also to decrease mode size. Among them, CSP1 is used for feature extraction, i.e., Backbone, while CSP2 is used for feature fusion, i.e., Neck. Third, Backbone not only has CSP1, but also CBS and SPPF modules. The SPPF module connects three MaxPool layers of 5 * 5 size in a series, passes the input through these three MaxPool layers in turn, and performs a Concat operation on the output of these three MaxPool before performing a CBS operation. The output of SPPF is the same as that of SPP, but SPPF runs faster. Fourth, Neck utilizes a path aggregation network (PANet) [19]. In PANet, low-level features are propagated using a new feature pyramid network (FPN) structure with enhanced bottom-up paths. In addition, adaptive feature pooling, which links feature grids to all feature levels, propagates useful information within each feature level directly to the next layer. As a result, PANet can improve the accuracy of the location of objects in lower layers by utilizing accurate localization signals. Finally, Output generates three

feature maps of different sizes, enabling the model to handle small, medium, and large objects for multi-scale [22] prediction.

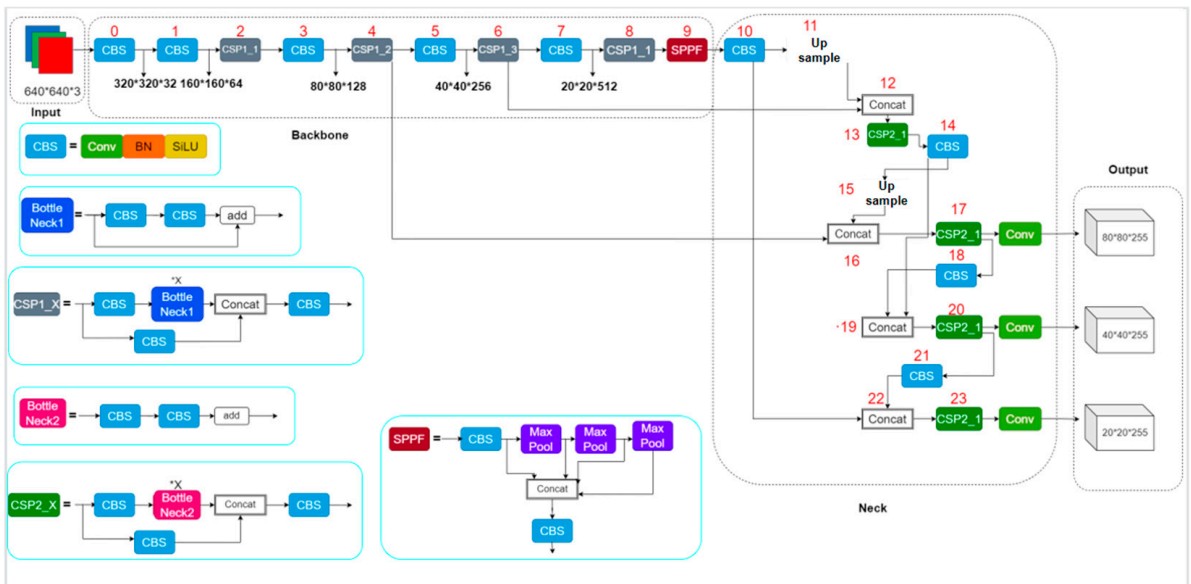

**Figure 3.** Model structure of YOLOv5s version 6.1.

### 2.3. SPPFP

SPPF is available as the last module of Backbone in YOLOv5 version 6.1. The SPPF module is a series of three MaxPool layers of $5 \times 5$ size through which inputs are then passed in turn, and a Concat operation is performed on the output of the three MaxPool layers before the CBS operation is performed. The structure of SPPF is shown in Figure 4. Maximum pooling and jump connection at different scales enable the image to learn features at different scales, and then fuse local and global features to enrich the representativeness of the feature map. Among them, maximum pooling divides the image into several rectangular regions outputting the maximum value for each sub-region. Although the maximum pooling operation can reduce the redundant information, it also tends to cause the loss of feature information.

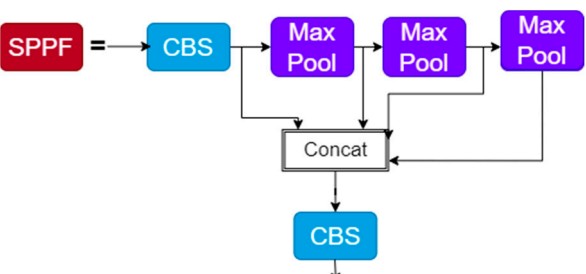

**Figure 4.** The structure of SPPF.

In this paper, the SPPF is improved by borrowing BenseNet's [23] construction of dense links and enhancing the idea of feature reuse. We then obtain the SPPF module, thereby reducing the loss of feature information caused by maximum module pooling. The SPPFP module is used to better retain global information on small-targets forest fires. Figure 5 shows the structure of SPPFP.

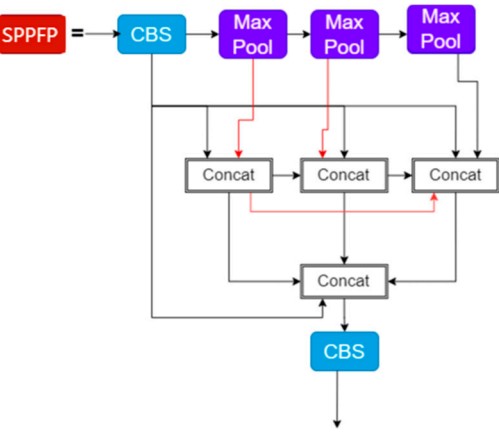

**Figure 5.** The structure of SPPFP.

*2.4. CBAM*

　　Due to the problem of small forest fire targets with low pixels in the picture, it is easy for missing information to occur. CBAM consists of a channel attention module and a spatial attention module in series. First, feature F is input into the channel attention module to get channel attention weight $M_C(F)$. Then, $M_C(F)$ is multiplied by bit with feature F to get feature $F'$. Next, the feature is input into the spatial attention module to get spatial attention weight $M_s(F')$. Finally, $M_s(F')$ is multiplied by bit with feature $F'$ to get feature $F''$. The structure of the CBAM attention module is shown in Figure 6.

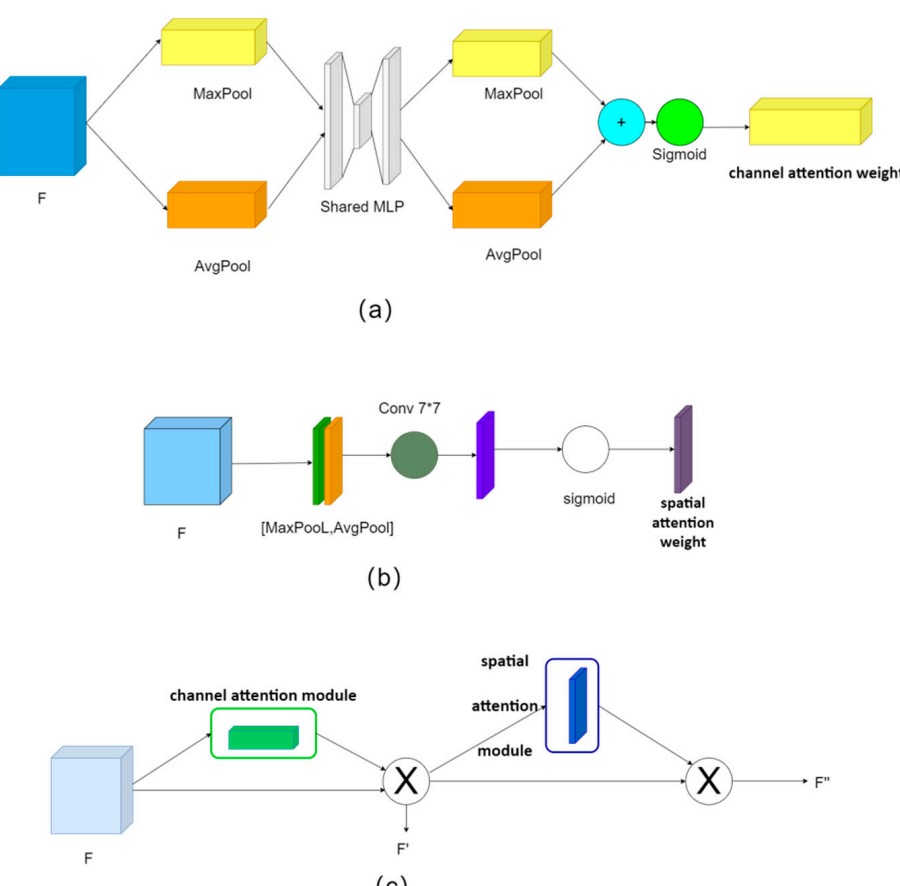

**Figure 6.** (**a**) The structure of the channel attention module. (**b**) The structure of the spatial attention module. (**c**) The structure of the CBAM attention module.

In the CBAM attention module, the channel attention module can utilize the information between feature channels, and the spatial attention module can utilize the information between feature spaces. The advantages of the two attention modules can be complemented to achieve effective attention to small targets. Adding the CBAM attention module to the Backbone of YOLOv5 can effectively improve the feature extraction ability and detection accuracy of the network. The CBAM attention module is added to YOLOv5 in the Backbone way as shown in Figure 7.

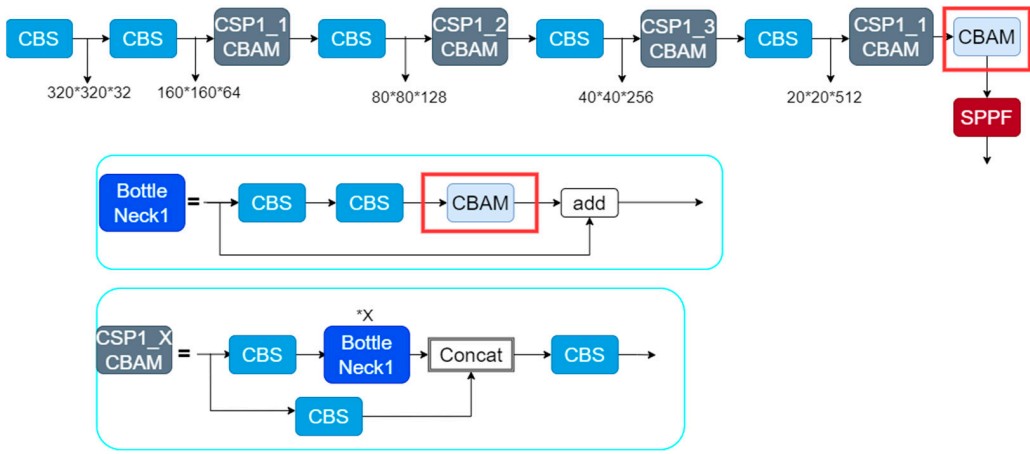

**Figure 7.** CBAM attention module added to YOLOv5 in the Backbone way.

In this paper, two types of additions are designed. In the first one, the CBAM attention module is added before the SPPF module; in the second one, the CBAM attention module is added before the add operation in BottleNeck1.

*2.5. BiFPN*

The full name of BiFPN is Bi-directional Feature Pyramid Network, which is a modified PANet structure. The structures of BiFPN and PANet are shown in Figure 8.

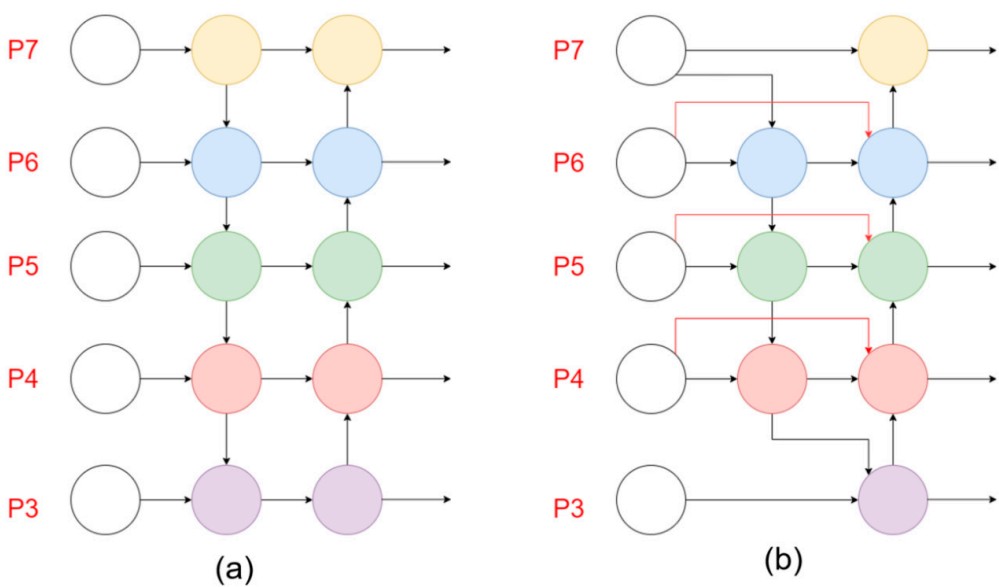

**Figure 8.** The structures of BiFPN and PANet. (**a**) PANet; (**b**) BiFPN.

Using BiFPN to improve YOLOv5's Neck allows for simpler and faster multi-scale feature fusion. Moreover, introducing learnable weights to Bi-FPN enables it to learn the importance of different input features and apply top-down and bottom-up multi-scale feature fusion repeatedly. Bi-FPN has better performance with fewer parameters and

FLOPS than Yolov5's Neck PANet. Due to this, it allows for better real-time forest fire detection.

### 2.6. Improved YOLOv5s Model

The original output layer of YOLOv5s version 6.1 generates only three feature maps of different sizes for processing large, medium, and small target objects. However, the targets of initial forest fires and long-distance shooting forest fires are very small, and the original three feature maps of different sizes are less effective in dealing with initial forest fires and long-distance shooting forest fires. Therefore, in this paper, one feature map is added to the original output layer of YOLOv5s version 6.1 for dealing with very small targets.

The model structure of the improved version 6.1 of YOLOv5s in this paper is shown in Figure 9.

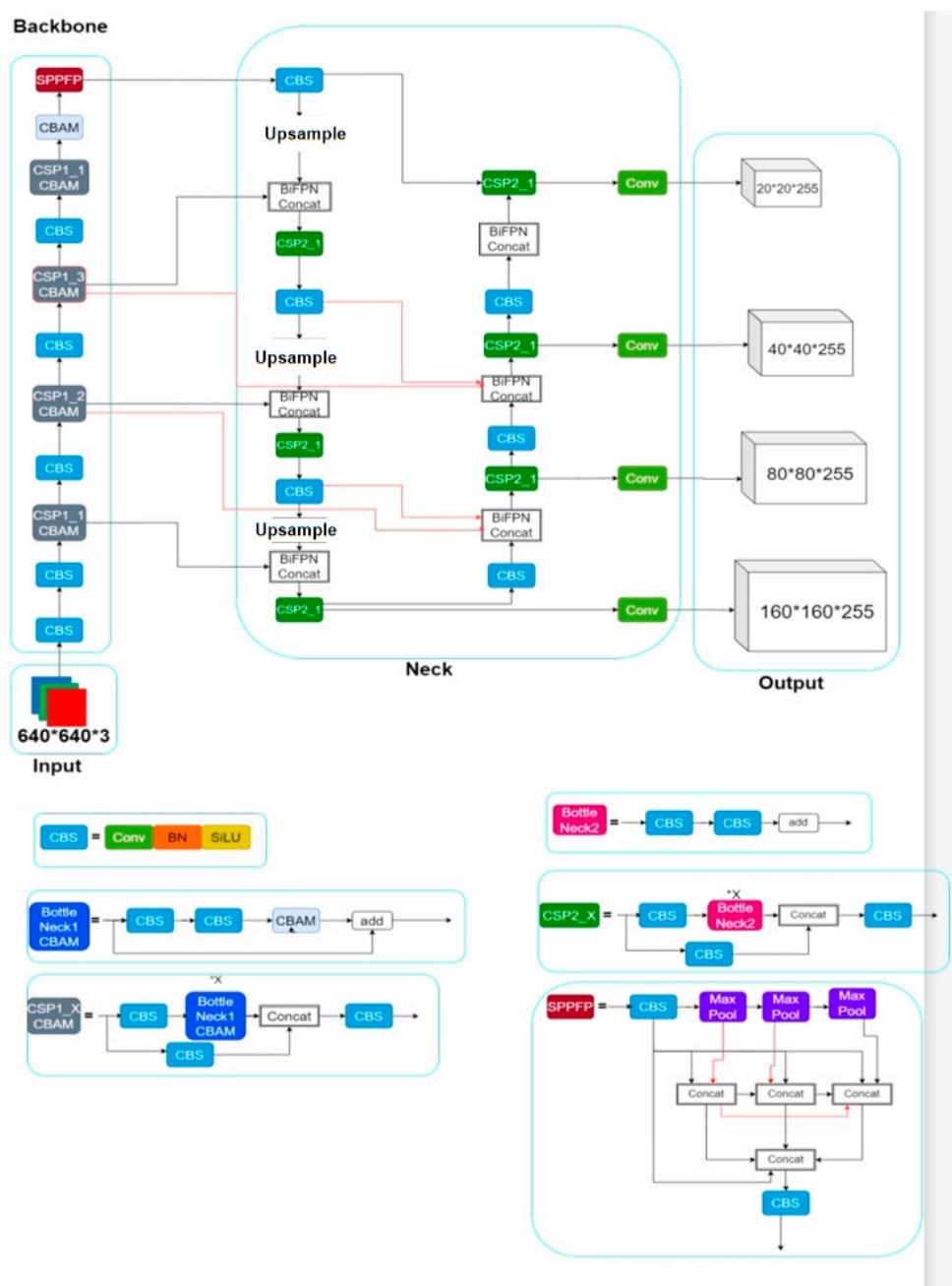

**Figure 9.** Improved version 6.1 of YOLOv5s.

*2.7. Transfer Learning*

　　The deep neural network model needs a large number of samples for training in order for the model to have a good effect. Since the initial fire data set belongs to a small sample data set, it is difficult to obtain a good detection effect by training directly from scratch.

　　Transfer learning is a method of applying already-obtained knowledge about a known domain to the target domain, fine-tuning is a method of making an entire pre-trained network on a known dataset to train the target dataset, using the already-trained model used as the initialized model on which the target dataset is trained.The trained model is used as the initial model, and the target data set is trained on this basis.

　　In order to improve the accuracy of the small-target forest fire detection model in this paper, the transfer learning method was used to train the small-target forest fire detection model. The forest fire dataset is trained in order to obtain the forest fire detection model, and then the knowledge is transferred to the forest fire detection model to train the small-target forest fire training set and thusly obtain the small-target forest fire detection model. Figure 10 shows the transfer learning process of training the small-target forest fire detection model.

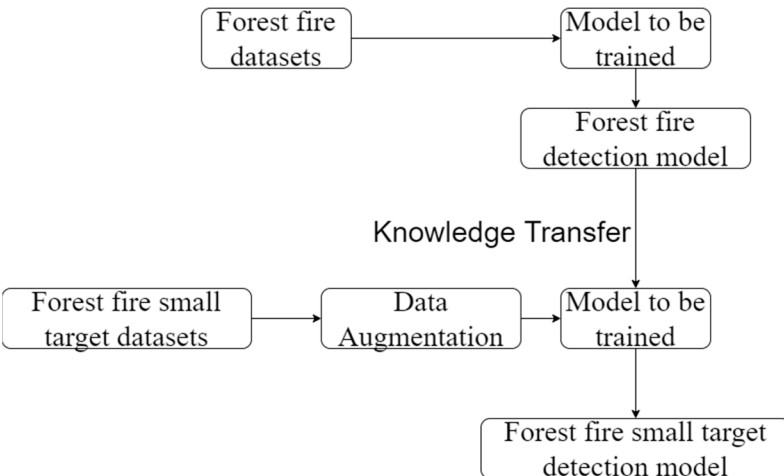

**Figure 10.** Transfer learning process of training the small-target forest fire detection model.

*2.8. Model Evaluation*

　　In this paper, we evaluate the model using the evaluation criterion of PASCAL VOC, one of the evaluation criteria that is widely used in target detection tasks. The evaluation metrics of PASCAL VOC are measured by the mean average precision when the IOU is set to 0.5, which is mAP@0.5. To calculate mAP, we need to use precision and recall, and the calculation formula is as follows.

$$\text{Precision} = \text{TP}/(\text{TP} + \text{FP}) \tag{1}$$

$$\text{Recall} = \text{TP}/(\text{TP} + \text{FN}) \tag{2}$$

$$\text{AP} = \int_0^1 P(r)\mathrm{d}r \tag{3}$$

$$\text{mAP} = \sum_{i=1}^{C} \text{AP}_i/C \tag{4}$$

　　In those equations, TP indicates that the target is a forest fire, and the network model detects the forest fire target. FP indicates that the target is not a forest fire, and the network model detects the non-forest fire target. FN indicates that the target is a forest fire, and the network model detects the non-forest fire target.

P(r) denotes a P–R curve with precision as the vertical coordinate and recall as the horizontal coordinate. AP is the area enclosed by precision and recall on a curve. The mAP is the average of the AP values for all categories.

## 3. Results

### 3.1. Training

The experimental conditions in this paper are shown in Table 1. The training parameter settings for the forest fire detection model are shown in Table 2. The training parameters for the small-target forest fire detection model are set as shown in Table 3. Both the forest fire dataset and the small-target forest fire dataset are divided into training set, validation set, and test set in 8:1:1. In addition, the small-target forest fire dataset was enhanced with data for the training set, validation set, and test set, respectively, after the allocation was completed. Details of the forest fire dataset and the small-target forest fire dataset are shown in Table 4.

**Table 1.** Experimental conditions.

| Experimental Environment | Details |
|---|---|
| Programming language | Python 3.9 |
| Operating system | Windows 10 |
| Deep learning framework | Pytorch 1.8.2 |
| GPU | NVIDIA GeForce GTX 1070 |

**Table 2.** Training parameters of the forest fire detection model.

| Training Parameters | Details |
|---|---|
| Epochs | 300 |
| batch-size | 16 |
| img-size (pixels) | $640 \times 640$ |
| Initial learning rate | 0.01 |
| Optimization algorithm | SGD |
| Pre-training weights file | None |

**Table 3.** Training parameters of the small target forest fire detection model.

| Training Parameters | Details |
|---|---|
| Epochs | 250 |
| batch-size | 16 |
| img-size (pixels) | $640 \times 640$ |
| Initial learning rate | 0.01 |
| Optimization algorithm | SGD |
| Pre-training weights | The best.pt obtained by training the corresponding forest fire detection model |

**Table 4.** Details of the two datasets.

| Dataset | Train | Val | Test |
|---|---|---|---|
| Forest fire dataset | 2537 | 282 | 314 |
| Small target forest fire dataset | 240 | 30 | 30 |

### 3.2. Ablation Experiments

The training process for each experiment is as follows. First, the forest fire detection model was obtained by training using the forest fire dataset and tested using the test set to obtain mAP@0.5. Then, the data of the forest fire detection model are used as pre-training weights for the training of the small-target forest fire model and are trained using the small-target forest fire dataset. Finally, the small-target forest fire detection model is tested using the test set in the small-target forest fire dataset to obtain mAP@0.5-S and FPS. The

data of the ablation experiments are shown in Table 5; VST denotes the very-small-target detection layer.

**Table 5.** The data of the ablation experiments.

| MODEL | Forest Fire Detection | Small Target Forest Fire Detection | |
|---|---|---|---|
| | mAP@0.5 | mAP@0.5-S | FPS |
| YOLOv5s | 76.1 | 60.2 | 55.2 |
| YOLOv5s + CBAM | 79.8 | 62.5 | 56.5 |
| YOLOv5s + SPPFP | 79.4 | 64.2 | 53.1 |
| YOLOv5s + BiFPN | 78.9 | 64.8 | 55.3 |
| YOLOv5s + VST | 79 | 64.7 | 51.5 |
| YOLOv5s + VST + CBAM | 79.9 | 65.7 | 51.7 |
| YOLOv5s + VST + CBAM + SPPFP | 81.1 | 66.1 | 52.1 |
| YOLOv5s + VST + CBAM + SPPFP + BiFPN | 82.1 | 70.3 | 54.1 |

*3.3. Comparison*

After the ablation experiments, we can see that, although YOLOv5 is one of the most advanced target detection models available, its mAP@0.5 and mAP@0.5-S are relatively low. In addition, the CBAM was added to YOLOv5s, the SPPF was replaced with SPPFP, the very-small-target detection layer was added and the Neck layer, and the PANet structure was replaced with the BiFPN structure in Experiments 2–5, respectively. In Experiment 2, mAP@0.5 and mAP@0.5-S improved by 3.7% and 2.3%. This demonstrates that adding the CBAM attention module to YOLOv5s can effectively improve the detection performance on small-target forest fires. In Experiment 3, mAP@0.5 and mAP@0.5-S improved by 3.3% and 4.0%. In addition, the FPS was even improved. These prove that the improved SPPF module can effectively improve the detection performance of the module. In Experiment 4, mAP@0.5 and mAP@0.5-S improved by 2.8% and 4.6%. This indicates that modifying the PANet structure in YOLOv5s to a BiFPN structure can effectively improve the detection performance of the module. In Experiment 5, mAP@0.5 and mAP@0.5-S were boosted by 2.9% and 4.5%. This demonstrates that adding a very-small-target detection layer can improve the performance of the model for detecting small-target forest fires. By comparing the data, we can conclude that all four changes improve the performance of the forest fire detection model and the small-target forest fire detection model to different degrees.

More importantly, these four changes were fused sequentially in Experiments 6–8. The mAP@0.5 and mAP@0.5-S of Experiment 6 are higher than those of Experiments 1, 2, and 5, and the FPS is also lower than the FPS of these three experiments. These evidence that adding both the very-small-target detection layer and the CBAM attention module can increase the detection performance of the module for small-target forest fires. Both of these improvements are effective improvements. Experiment 7 has higher mAP@0.5 and mAP@0.5-S than Experiments 1, 2, 3, and 5, and the mAP@0.5 and mAP@0.5-S of Experiment 7 are higher compared to Experiment 6. These prove that fusing the very-small-target detection layer, CBAM attention module, and SPPFP module together can effectively improve the detection performance of the model for small-target forest fires. Experiment 8 has higher a and b than the previous Experiments 1–7. We can conclude that the improved YOLOv5s structure proposed in this paper possesses better performance compared to YOLOv5s. mAP@0.5 showed a 6.1% improvement compared to baseline; mAP@0.5-S showed a 10.1% improvement compared to baseline.

After this comprehensive improvement, the small-target forest fire detection model has better detection performance for small-target forest fires, and facing a target like a forest fire, our model suffers less interference (Figures 11 and 12).

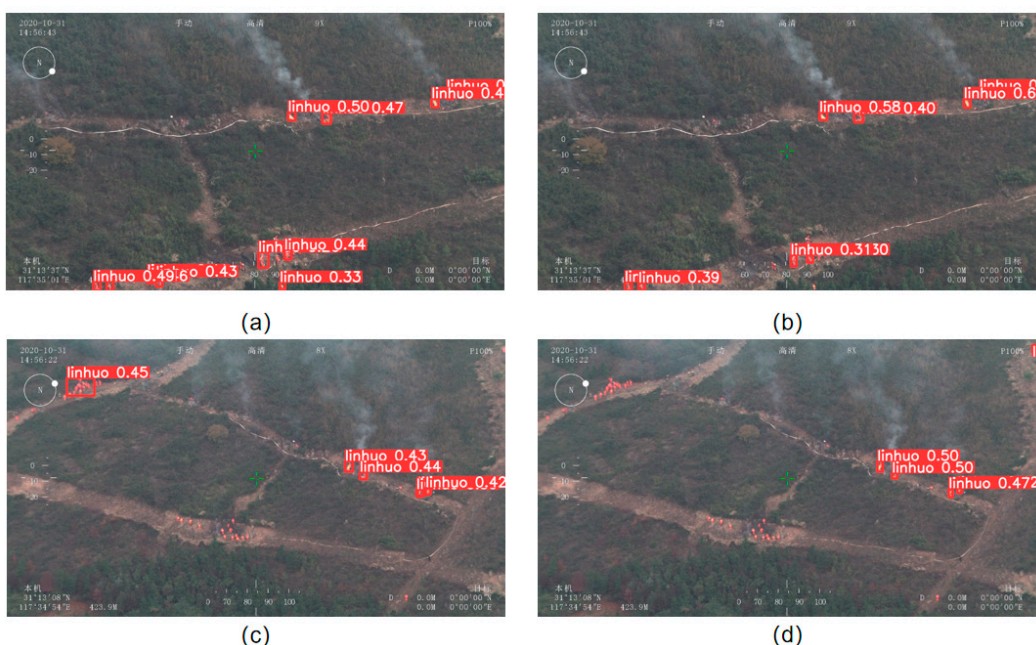

**Figure 11.** Comparison of YOLOv5s and our detection model. Our small-target forest fire detection model has better performance. (**a**) Four small forest fire targets were detected by YOLOv5s, with seven detection errors. (**b**) Four small forest fire targets were detected by our model, with four detection errors. (**c**) Two small forest fire targets were detected by YOLOv5s, with three detection errors. (**d**) Two small forest fire targets were detected by YOLOv5s, with two detection errors.

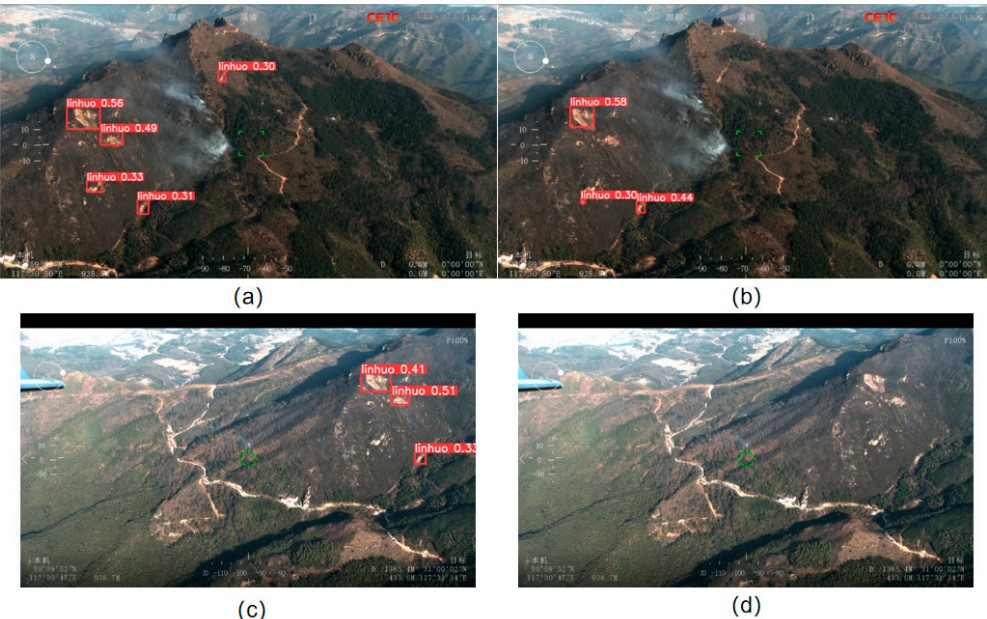

**Figure 12.** Comparison of YOLOv5s and our model. Our model suffers less interference. (**a**) Five detection errors detected by YOLOv5s. (**b**) Three detection errors detected by our model. (**c**) Three detection errors detected by YOLOv5s. (**d**) No error detections detected by our model.

## 4. Discussion

Compared to common objects in fixed form such as chairs, tables, doors, etc., forest fires are dynamic objects [24]. Many large forest fires have developed from small forest fires. If small forest fires are not detected and acted upon immediately, they can lead to large forest fires and have very serious consequences. Therefore, it is important to study the detection of early small forest fires. In addition, forest fires have a variety of shapes,

sizes, textures, and colors, making forest fire targets very complex. Furthermore, the small size and low pixel size of a forest fire target from a long distance can cause serious missing of forest fire target features. For the above reasons, it is very difficult to detect small-target forest fires photographed at a distance.

Therefore, it is important to improve the performance of the detector to identify small-target forest fires. Through experiments, it was found that YOLOv5 has good detection performance for large and medium-sized forest fires, but not for small forest fires. YOLOv5 often misses small-target forest fires. Therefore, this paper improves the detection performance of YOLOv5 by adding a very-small-target detection layer, a CBAM attention module, and improving the original SPPF feature extraction module. However, these three operations lead to an increase in the number of parameters of the model, resulting in a decrease in the FPS of the model. This eventually leads to a decrease in the real-time performance of the small-target forest fire detection model.

In order to solve the problem of the decreasing real-time performance of model detection, we modify the original PANet structure of the Neck layer of YOLOv5 into a BiFPN structure. BiFPN not only has fewer parameters and FLOPS compared to PANet, but also has a much better performance. Therefore, after optimizing the PANet structure to a BiFPN structure in the experiment, the model not only has better performance but also better real-time performance.

However, the small-target forest fire detection model in this paper still has shortcomings. Therefore, we will continue to optimize this small-target forest fire detection model. First, we will research an annotation strategy for the forest fire dataset and use this strategy to help the model to better extract the features of the target. As the performance of the model is very much related to the good or bad labeling of the dataset, a well-labeled dataset can greatly improve the detection performance of the model. Second, we will examine other ways to improve the performance of the model and reduce the likelihood of the model being disturbed by non-forest fire targets. Last but not least, in order to have better deployment of the model, we will also study the light-weighting of the model and improve the real-time detection performance of the model.

The experimental results show that the model proposed in this paper has good prospects for practical application. Since forests are mostly found in remote areas, the cameras are mounted on UAVs or helicopters in practical applications. However the model proposed in this paper can generate false alarms due to the pixels of the camera and fire-like objects. That said, compared with forest fire detection methods that lay a large number of sensors in the forest, the method proposed in this paper not only saves cost, but also provides firefighters with on-the-spot fire reference. The use of camera images not only has better real-time functioning compared to the use of satellite remote sensing images, but it can also detect initial forest fires and thereby reduce damage. Therefore, we think that the method proposed in this paper may have more advantages.

The small-target forest fire identification model proposed in this paper requires cameras as sensors in practical deployment. Although good real-time performance can be obtained by using a camera as a sensor, it is susceptible to factors such as light and occlusions. The model in this paper has been improved to reduce the false alarm rate, but there are still false alarms. We will continue to investigate how to reduce the false alarm rate and further improve the detection performance of the model in subsequent research.

In the follow-up study, we will deploy different cameras on the UAV for forest fire detection experiments. Camera types include 360-degree panoramic cameras, conventional HD cameras, and OAK cameras. In addition, we note that Sengan et al. [25] proposes a real-time automatic survey of Indian road animals by using deep learning for 3D reconstruction detection. This paper uses deep learning for deep-learning real 3D motion-based YOLOv3 (R-3D-YOLOV3) image classification and filtering. This article provides some inspiration for our subsequent work. Last but not least, we note that Zhang et al. [26] proposed a forest fire detection system based on acoustic sensors. In the simulation experiments, the recognition rate of the method can reach about 70%. The use of acoustic sensors can detect

early forest fires that cannot be detected using cameras. It is hoped that the advantages of multiple forest fire detection methods can be combined with each other in future research, and that the accuracy of forest fire detection will be greatly improved.

## 5. Conclusions

Many of the large forest fires that occur globally have evolved from small forest fires. In addition, since forest fires are dynamic targets, it is impossible to have good performance using conventional target detection models. Therefore, it is important to study how to detect initial forest fires and small-target forest fires.

To address these problems, a small-target forest fire detection model is proposed in this paper. First, the CBAM attention module and the very-small-target detection layer are added to YOLOv5 to enhance the model's attention to small targets of forest fires. Second, the closely connected structure is used to improve the SPPF module and reduce the loss of feature information due to maximum pooling in the SPPF module. Finally, optimizing PANet to BiFPN allows the model to have better performance with fewer parameters.

The experimental results show that the performance of our model is significantly improved compared to YOLOv5s, which makes the model promising for small-target forest fire detection. In subsequent research work, we will proceed to test the small-target forest fire detection module proposed in this paper for practical applications.

**Author Contributions:** Z.X. devised the programs and drafted the initial manuscript. H.L. and F.W. designed the project and revised the manuscript. All authors have read and agreed to the published version of the manuscript.

**Funding:** This work was supported by the Key Research and Development plan of Jiangsu Province (Grant No. BE2021716), the Jiangsu Modern Agricultural Machinery Equipment and Technology Demonstration and Promotion Project (NJ2021-19), the National Natural Science Foundation of China (32101535), and the Jiangsu Postdoctoral Research Foundation (2021K112B).

**Institutional Review Board Statement:** Not applicable.

**Informed Consent Statement:** Not applicable.

**Conflicts of Interest:** The authors declare no conflict of interest.

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
