# Peer review of "A Small Target Forest Fire Detection Model Based on YOLOv5 Improvement"

_forests, doi:10.3390/f13081332_

Round 1

Reviewer 1 Report

Dear Authors!

Thanks for the opportunity to get acquainted with your really interesting and state-of-the art research! I hope that my notes will be useful and contribute to the improvement of your manuscript!

The aim of the study is to develop and propose a detection model for forest fires with small size. The need for innovation to increase efficiency in detecting, monitoring and fighting forest fires is indisputable, and in this regard, research such as the one proposed is useful and necessary. In order to achieve the aim of the study, authors have developed an improved version of YOLOv5 model, which is based on object detection. As training sets, authors have used selected and suitable images of forest fires from the web and from some public forest fire datasets. The suggested improved structure has been evaluated using the evaluation criterion of PASCAL VOC and the results have been compared with the obtained by the traditional YOLOv5 model, showing better performance with about 10%. However, as the main drawback, I would note the presence of many terms mentioned only as abbreviations, without a more detailed description of what exactly is behind them. Of course, the article is written at a high professional level and obviously targets narrow specialists as an audience, but to make it more readable and understandable by a wider range of readers, I suggest adding 1-2 clarifying sentences for the mentioned models, modules, layers etc., especially in the Introduction. In addition, it is not clear how the proposed model can be applied in practice? What is needed to be implemented? As far as I understood, the model is a detection model based on cameras as detectors. In the paper, It has not been specified what sensors are used. If the sensors are cameras, should these cameras be put somewhere or the model would rely on available ones? Forests are remote sites and this would be a problem. Certainly, the proposed solution has its drawbacks. However, such are not indicated in the article. On the other side, some of the disadvantages of other monitoring systems and methods are mentioned, as for example those of satellite remote sensing. But how, the proposed solution is better? I suggest expanding the Discussion in this direction!

Kind regards!

Author Response

Reviewer #1:

The aim of the study is to develop and propose a detection model for forest fires with small size. The need for innovation to increase efficiency in detecting, monitoring and fighting forest fires is indisputable, and in this regard, research such as the one proposed is useful and necessary. In order to achieve the aim of the study, authors have developed an improved version of YOLOv5 model, which is based on object detection. As training sets, authors have used selected and suitable images of forest fires from the web and from some public forest fire datasets. The suggested improved structure has been evaluated using the evaluation criterion of PASCAL VOC and the results have been compared with the obtained by the traditional YOLOv5 model, showing better performance with about 10%. However, as the main drawback, I would note the presence of many terms mentioned only as abbreviations, without a more detailed description of what exactly is behind them. Of course, the article is written at a high professional level and obviously targets narrow specialists as an audience, but to make it more readable and understandable by a wider range of readers, I suggest adding 1-2 clarifying sentences for the mentioned models, modules, layers etc., especially in the Introduction. In addition, it is not clear how the proposed model can be applied in practice? What is needed to be implemented? As far as I understood, the model is a detection model based on cameras as detectors. In the paper, It has not been specified what sensors are used. If the sensors are cameras, should these cameras be put somewhere or the model would rely on available ones? Forests are remote sites and this would be a problem. Certainly, the proposed solution has its drawbacks. However, such are not indicated in the article. On the other side, some of the disadvantages of other monitoring systems and methods are mentioned, as for example those of satellite remote sensing. But how, the proposed solution is better? I suggest expanding the Discussion in this direction!

  1. The presence of many terms mentioned only as abbreviations, without a more detailed description of what exactly is behind them.

Reply to Question 1: This is a very helpful question and we have made necessary changes. Firstly, in the abstract section, we have added full names for terms that were originally mentioned as abbreviations.

Changes Made: According to this comment, we have made necessary changes, which is colored blue in the revised manuscript.

  1. It is not clear how the proposed model can be applied in practice? What is needed to be implemented? As far as I understood, the model is a detection model based on cameras as detectors. In the paper, It has not been specified what sensors are used. If the sensors are cameras, should these cameras be put somewhere or the model would rely on available ones? Forests are remote sites and this would be a problem.

Reply to Question 2: These are good question and we have made necessary changes. Firstly, in the abstract section, we add that the module proposed in this paper requires  cameras as sensors in its application. Secondly, in the discussion section, we add a more detailed discussion of the prospects for practical applications of the model proposed in this paper.

Changes Made: According to this comment, we have made necessary changes, which is colored blue in the revised manuscript.

  1. Certainly, the proposed solution has its drawbacks. However, such are not indicated in the article. On the other side, some of the disadvantages of other monitoring systems and methods are mentioned, as for example those of satellite remote sensing. But how, the proposed solution is better? I suggest expanding the Discussion in this direction!

Reply to Question 3: This is a very good suggestion and we have made necessary changes. We have added in the introduction section not only the advantages of forest fire detection using cameras as sensors compared to satellite images, but also the advantages of forest fire detection using cameras as sensors compared to conventional sensors that do not provide image information. In addition, We have added a comparison of forest fire detection using cameras as sensors and other monitoring systems and methods in the discussion section.

Changes Made: According to this comment, we have made necessary changes, which is colored blue in the revised manuscript.

Reviewer 2 Report

The authors propose a small target wildfire detection model, using a CBAM attention module and the very small target detection layer are added to YOLOv5 to improve the model attention to small wildfire targets and optimize PANet a BiFPN allows the model to perform better with fewer parameters.

Specific comments

Adapt the references according to the guide for authors

Line 29. What does "identification" refer to, or does it refer to early fire detection, explain for greater clarity.

LIne 36. Describe what kind of sensors.

Line 41. What happens with high spatial resolution sensors, is it the same? please clarify.

Line 41-43. What happens with high spatial resolution sensors, is it the same? In addition to satellite images that are free, their type of spectral and radiometric resolution, please clarify.

Line 41-43. What happens with high spatial resolution sensors, is it the same? In addition to satellite images that are free, their type of spectral and radiometric resolution, please clarify.

Line 108. It would be interesting to know where the photos were taken.

Line 122. Add a space.

Line 202. Improve the quality of the figure

Line 262. Add a space.

Line 272-273. Table 2 and 3. In what units is the image size (img-size)

Line 273. Add a space to separate the tables.

Line. 321. Mention which objects, be clearer in the writing.

Line 322. As long as they are not detected in time, it is important to point it out, since this gives importance to this study.

Line 324. Writing

Line 324. Throughout the document, the small size is not defined, which proportion has been considered as a small fire, in surface or image size, this is very important since it marks the limitation of YOLOv5 to determine possible recommendations of its application.

Author Response

The authors propose a small target wildfire detection model, using a CBAM attention module and the very small target detection layer are added to YOLOv5 to improve the model attention to small wildfire targets and optimize PANet a BiFPN allows the model to perform better with fewer parameters.

  1. Adapt the references according to the guide for authors

Reply to Question 1: This is a good question and we have made necessary changes.We checked the format of the literature and corrected the names of incorrectly formatted documents. In addition, we added several pieces of literature related to the study.

Changes Made: According to this comment, we have made necessary changes, which is colored blue in the revised manuscript.

  1. Line 29. What does "identification" refer to, or does it refer to early fire detection, explain for greater clarity.

Reply to Question 2: This is a good question and we have made necessary changes. It was an inadvertent oversight on our part not to give a clear definition here, and we have changed this to initial forest fire detection.

Changes Made: According to this comment, we have made necessary changes, which is colored blue in the revised manuscript.

  1. LIne 36. Describe what kind of sensors.

Reply to Question 3: This is a good question and we have made necessary changes. We have added some of the sensor types used for forest fire detection.

Changes Made: According to this comment, we have made necessary changes, which is colored blue in the revised manuscript.

  1. Line 41-43. What happens with high spatial resolution sensors, is it the same? In addition to satellite images that are free, their type of spectral and radiometric resolution, please clarify.

Reply to Question 4: This is a good question and we have made necessary changes. We reviewed the relevant literature and added the advantages and disadvantages of applying high spatial resolution sensors to forest fire identification.

Changes Made: According to this comment, we have made necessary changes, which is colored blue in the revised manuscript.

  1. Line 108. It would be interesting to know where the photos were taken.

Reply to Question 5: This is a good question. However, we use images from the dataset by manually integrating images from existing databases and images crawled from the Internet. So we do not really know where the photos were taken. We are very sorry for this.

  1. Line 122. Add a space.Line 262. Add a space.Line 273. Add a space to separate the tables.

Reply to Question 6:There are good questions and we have made necessary changes.

Changes Made: According to these comments, we have made necessary changes.

  1. Line 202. Improve the quality of the figure

Reply to Question 7: This is a good question and we have made necessary changes. We re-drew Figure 8 with drawing software to make its picture of higher quality.

Changes Made: According to this comment, we have made necessary changes, which is colored blue in the revised manuscript.

  1. Line 272-273. Table 2 and 3. In what units is the image size (img-size)

Reply to Question 8: This is a good question and we have made necessary changes. We have not only added the image size (img-size) in pixels, but also modified the format of writing the image size (img-size) data.

Changes Made: According to this comment, we have made necessary changes, which is colored blue in the revised manuscript.

  1. 321. Mention which objects, be clearer in the writing.

Reply to Question 8: This is a good question and we have made necessary changes. We have given examples of conventional fixed objects so that the reader can better understand the difference between conventional fixed objects and forest fires.

Changes Made: According to this comment, we have made necessary changes, which is colored blue in the revised manuscript.

  1. As long as they are not detected in time, it is important to point it out, since this gives importance to this study.

Reply to Question 8: This is a good question and we have made necessary changes. Originally we only wrote that large forest fires all develop from small forest fires, but forgot to highlight the importance of studying the identification of small target forest fires. We have added and highlighted the importance of studying small target forest fire identification here.

Changes Made: According to this comment, we have made necessary changes, which is colored blue in the revised manuscript.

  1. Line 324. Throughout the document, the small size is not defined, which proportion has been considered as a small fire, in surface or image size, this is very important since it marks the limitation of YOLOv5 to determine possible recommendations of its application.

Reply to Question 8: This is a good question and we have made necessary changes.Due to an oversight on our part, we forgot to indicate the specific criteria for judging small targets in this document. We have added specific criteria for judging small targets in section 2.1.

Changes Made: According to this comment, we have made necessary changes, which is colored blue in the revised manuscript.

Reviewer 3 Report

The authors present A Small Target Forest Fire Detection Model Based on YOLOv5 Improvements, which is a hot research topic of the recent times.

Some suggestions to improve the quality of the paper:

Do you see any potential of using 360 Degree Videos in your research? And mention the suitable camera for the detection of a bee?

 Authors should explain the reason why they choose these algorithms. What are the limitations of this work? How can the rigor of this work be demonstrated?

Sufficient information about the previous study findings is presented for readers to follow the present study rationale and procedures.

  Results need more explanations. Additional analysis is required at each experiment to show the its main purpose.

I want to make special mention that figures are pixelated in bad way gives no good impression to the readers. Authors are suggested to replace all images with good quality ones.

Recent references may be included, Unmanned Aerial Vehicle (UAV) based Forest Fire Detection and monitoring for reducing false alarms in forest-fires  and Real-time automatic investigation of indian roadway animals by 3D reconstruction detection using deep learning for R-3D-YOLOV3 image classification and filtering

     The writing of the paper needs a lot of improvement in terms of grammar, spellings and presentations. The paper needs a         careful English polishing since there are many typos and poorly written sentences.

 Considering all of the above, I recommend the paper to be revised carefully.

Author Response

Reviewer #3:

The authors present A Small Target Forest Fire Detection Model Based on YOLOv5 Improvements, which is a hot research topic of the recent times.

Some suggestions to improve the quality of the paper:

Do you see any potential of using 360 Degree Videos in your research? And mention the suitable camera for the detection of a bee?

 Authors should explain the reason why they choose these algorithms. What are the limitations of this work? How can the rigor of this work be demonstrated?

Sufficient information about the previous study findings is presented for readers to follow the present study rationale and procedures.

  Results need more explanations. Additional analysis is required at each experiment to show the its main purpose.

I want to make special mention that figures are pixelated in bad way gives no good impression to the readers. Authors are suggested to replace all images with good quality ones.

Recent references may be included, Unmanned Aerial Vehicle (UAV) based Forest Fire Detection and monitoring for reducing false alarms in forest-fires  and Real-time automatic investigation of indian roadway animals by 3D reconstruction detection using deep learning for R-3D-YOLOV3 image classification and filtering

     The writing of the paper needs a lot of improvement in terms of grammar, spellings and presentations. The paper needs a careful English polishing since there are many typos and poorly written sentences.

 Considering all of the above, I recommend the paper to be revised carefully.

  1. Do you see any potential of using 360 Degree Videos in your research? And mention the suitable camera for the detection of a bee?

Reply to Question 1: This is a good question and we have made necessary changes. In the Discussion section, we add an outlook on the types of cameras to be used for subsequent practical deployments, and intend to test the effects of deploying different cameras on UAVs in subsequent experiments.

Changes Made: According to this comment, we have made necessary changes, which is colored blue in the revised manuscript.

  1. Authors should explain the reason why they choose these algorithms. What are the limitations of this work? How can the rigor of this work be demonstrated?

Reply to Question 2: These are good questions and we have made necessary changes. Firstly, we added in 2.2 why we chose YOLOv5s. And we explained in sections 2.3, 2.4, and 2.5 why we made these optimizations. Secondly, we added the drawbacks of this work in the discussion section. Thirdly, the rigor of the experimental work is demonstrated by ablation experiments. We enriched the interpretation of the experimental results.

Changes Made: According to these comments, we have made necessary changes, which is colored blue in the revised manuscript.

  1. Results need more explanations. Additional analysis is required at each experiment to show the its main purpose.

Reply to Question 3: This is a good question and we have made necessary changes. We have enriched the interpretation of the experimental results so that the reader can understand the purpose of the experiment in more detail and easily.

Changes Made: According to these comments, we have made necessary changes, which is colored blue in the revised manuscript.

  1. I want to make special mention that figures are pixelated in bad way gives no good impression to the readers. Authors are suggested to replace all images with good quality ones.

Reply to Question 4: This is a good question and we have made necessary changes.We have modified Figures 6, 8, and 9. These three images are much clearer than before. It can give readers a good reading experience.

Changes Made: According to this comment, we have made necessary changes, which is colored blue in the revised manuscript.

  1. Recent references may be included, Unmanned Aerial Vehicle (UAV) based Forest Fire Detection and monitoring for reducing false alarms in forest-fires  and Real-time automatic investigation of indian roadway animals by 3D reconstruction detection using deep learning for R-3D-YOLOV3 image classification and filtering.

Reply to Question 5: This is a good question and we have made necessary changes. Thank you very much for providing us with two very good papers. We have added these two papers to our application literature.

Changes Made: According to this comment, we have made necessary changes, which is colored blue in the revised manuscript.

  1. The writing of the paper needs a lot of improvement in terms of grammar, spellings and presentations. The paper needs a careful English polishing since there are many typos and poorly written sentences.

Reply to Question 6: This is a good question and we have made necessary changes.We checked the paper for spelling mistakes and grammatical errors as far as we could.

Changes Made: According to this comment, we have made necessary changes, which is colored blue in the revised manuscript.

Round 2

Reviewer 2 Report

Thanks for the answers

Reviewer 3 Report

All the comments are addressed and I recommend the revised paper for publication.